# The Effectiveness of the Third Dose of COVID-19 Vaccine: When Should It Be Performed?

**DOI:** 10.3390/vaccines12030315

**Published:** 2024-03-16

**Authors:** Giacomo Biganzoli, Marco Mendola, Pier Mario Perrone, Laura Maria Antonangeli, Anna Beatrice Elena Longo, Paolo Carrer, Claudio Colosio, Dario Consonni, Giuseppe Marano, Patrizia Boracchi, Elia Biganzoli, Silvana Castaldi

**Affiliations:** 1Department of Biomedical and Clinical Sciences “L. Sacco”, University of Milan, 20157 Milan, Italy; giacomo.biganzoli@unimi.it (G.B.); paolo.carrer@unimi.it (P.C.); giuseppe.marano@unimi.it (G.M.); patrizia.boracchi@unimi.it (P.B.); elia.biganzoli@unimi.it (E.B.); 2Occupational Health Unit, Fatebenefratelli-Sacco University Hospital, 20157 Milan, Italy; mendola.marco@asst-fbf-sacco.it; 3Department of Biomedical Sciences for Health, University of Milan, 20133 Milan, Italy; silvana.castaldi@unimi.it; 4Department of Clinical Sciences and Community Health, University of Milan, 20122 Milan, Italy; 5Post Graduate School in Occupational Health, University of Milano, 20142 Milan, Italy; laura.antonangeli@unimi.it (L.M.A.); anna.longo@unimi.it (A.B.E.L.); 6Department of Health Sciences, University of Milano, 20142 Milan, Italy; claudio.colosio@unimi.it; 7Occupational Health Unit, Saints Paolo and Carlo University Hospital, 20142 Milan, Italy; 8Epidemiology Unit, Fondazione IRCCS Ca’ Granda Ospedale Maggiore Policlinico, 20122 Milan, Italy; dario.consonni@policlinico.mi.it; 9Quality Unit Fondazione IRCCS Ca’ Granda Ospedale Maggiore Policlinico, 20122 Milan, Italy

**Keywords:** COVID-19 vaccine, vaccine effectiveness, healthcare workers

## Abstract

Background: COVID-19 vaccination is the most significant step toward the long-term mitigation of SARS-CoV-2-related complication, avoiding disease and death and decreasing virus spread. This study aimed to evaluate, in a real-world setting, booster dose effectiveness to reduce COVID-19 risk considering the amount of time after the end of the two-dose vaccination cycle. A sub-analysis was conducted to adjust the booster dose effect for occupational and demographic factors. Methods: About 16,000 COVID-19-vaccinated HCWs of three University Hospital Networks in Milan (HN1/HN2/HN3) were included in the study. Data were collected by Occupational Health Physicians of the HNs within specific computerized databases. Results: In univariable analysis, booster dose administration displayed a slightly higher risk of infection with respect to not receiving it, OR = 1.18, with 95% confidence interval (C.I) [0.99, 1.41]. When the model was adjusted with the modulating effect of time from the completion of the vaccination cycle on booster dose administration, the latter resulted in strong protective effect against infection, OR = 0.43, 95% CI [0.26, 0.74]. However, considering the modifying influence of time from the vaccination cycle’s completion, the administration of booster doses appeared to have a protective effect against infection. In HN1, students and resident physicians displayed lower odds of infection with respect to physicians. Lastly, a non-linear effect of age was reported. Conclusions: Our findings suggest that the correct timing in vaccine scheduling and administration is critical to vaccine effectiveness. These findings, applicable to all vaccinations, should help in setting up more effective vaccination strategies.

## 1. Introduction

Starting in the first months of 2020, a new public health challenge due to Severe Acute Respiratory Syndrome Coronavirus 2 (SARS-CoV-2) caused a pandemic disease called Coronavirus Disease 19 (COVID-19) that, by 29 June 2023, resulted in 767,518,723 confirmed cases and 6,947,192 cumulative deaths [1]. Despite the recent appearance of this novel Coronavirus, the exceptional attention afforded to the situation by several nations and pharmaceutical companies allowed them to quickly develop many vaccines based on different technologies [2]. The presence of several vaccines also offers the possibility to organize mass vaccination campaigns considered to be one of the best public health strategies that are also cost effective [3,4,5]. Indeed, mass vaccination is considered the most important intervention to achieve sustained mitigation of the threat caused by COVID-19 since it prevents morbidity and mortality and reduces SARS-CoV-2 transmission [6]. On 27 December 2020, “Vaccine day” was held throughout Italy as well as throughout Europe, marking the “symbolic” start of the COVID-19 vaccination campaign. The nationwide vaccination campaign was therefore performed, initially aimed at higher-risk patient categories, including Healthcare Workers (HCWs), and the BNT162b2 mRNA COVID-19 vaccine was chosen for the vaccination of these categories by the Italian Minister of Health and administered according to a two-dose schedule (21-day interval) [7]. During the last month of 2021, a third-dose vaccination campaign started in Italy following the Minister of Health’s indications; also in this case, HCWs were included among the categories to which vaccination should be addressed as a priority [8]. Due to their importance in taking care of infected patients, HCWs were and are often considered the first category of candidates for vaccination in several countries, following international indications made by World Health Organization (WHO) and International Labor Organization (ILO) [9]. This is highlighted considering the high prevalence of HCW COVID-19 infections described in the literature, probably due to the strict and continuous contact between HCWs and COVID-19 patients [10,11]. The fear of vaccination side effects at the third dose as well as the idea that the booster dose does not provide additional protection are high, and some studies have reported an increased risk of side effects after subsequent doses [12,13,14]. Some investigations have also assessed this vaccination hesitancy among HCWs refusing third doses, considering lesser confidence in COVID-19 vaccines, misinformation on COVID-19 infection, and immune response to previous infection as the main reasons for vaccine refusal [12,13,14].

This study first aims at evaluating the potential effectiveness of booster dose administration to HCWs in a real-life setting after the completion of a two-dose vaccination-cycle completion. Data from registries of the main three different hospitals of the city of Milan were analyzed considering a follow-up period of 7 months from the start of the booster vaccination campaign.

## 2. Materials and Methods

### 2.1. Study Population

Our case series included all COVID-19-vaccinated HCWs of three large University Teaching Hospital networks in Milan (identified in the text as HN1, HN2, HN3) employing about 16,000 HCWs. All the considered hospitals’ networks have an agreement with the University of Milan to teach students of medicine and other health professionals and resident doctors.

Since the beginning of the pandemic, the Occupational Health Units of all involved hospitals have carried out intense activity in contact tracing and management of SARS-CoV-2 positive HCWs, following the national and regional legislative directives. Furthermore, from 28 December 2020, the physicians of the Occupational Units, supported by resident medical doctors, have also been actively involved in anti-COVID-19 vaccination. Vaccinations were performed using the BNT162b2 Pfizer–BioNTech© COVID-19 vaccine through the creation of an on-site vaccination center in the hospitals or sending a HCW to an external vaccine hub to ensure the utmost convenience and maximum adhesion. The vaccine was administered according to an initial two-dose schedule (21-day interval) and, approximately 8–10 months later, a subsequent booster dose. Therefore, booster doses were administrated between November 2021 and January 2022.

At the same time, the diagnosis of COVID-19 infection in HCWs was carried out by Occupational Health Units through the execution of PCR carried out on nasopharyngeal swabs in patients undergoing screening checks, in the presence of symptoms suggestive of infection or close contact with people with a confirmed infection in all three hospital networks assessed. After the administration of the third dose, no specific record about viral isolate variants was performed due to the high costs of analysis and the lack of specific utility in regard to the absence of differences in the treatment or management of the different variants.

### 2.2. Data Collection

Data related to the date of vaccinations and the number of doses administered within the three involved hospital networks were collected by Occupational Health Units within a specific computerized database. Moreover, demographic data were also collected such as gender, age, and anamnestic features with leading influences on vaccination. For those vaccinated elsewhere, information was completed by linking the cohort file with a regional vaccination database using Tax Code as a unique identifier. The specific computerized database in use at the Occupational Health Units also collected the number of nasal swabs performed with related results. Due to previous organization, HN2 and HN3 also collected clinical reports about signs and symptoms during COVID-19 infection to assess change in clinical findings among HCW with different vaccination doses. HN3 additionally assessed the hypothetical causes of infection such as close contact or professional exposition in a high-risk clinical ward. At the same time, HN1 collected professional features of HCWs infected by SARS-CoV-2 after vaccination, work position, and workplace.

### 2.3. Statistical Analysis

Since the measure of a real-life effectiveness of the booster dose administration was the primary endpoint of the study, this was evaluated by means of a time-to-event analysis. Here, the event of interest was SARS-CoV-2 infection. Namely, HCWs were observed in a period starting from the first day of the booster dose vaccination campaign to 7 months after that, when the follow-up administratively ended. During this period, HCWs could develop SARS-CoV-2 infection before or after booster dose administration or remain free of infection during the overall follow-up period. The latter were considered censored observations. 

Given the presence of the censored observations and the follow-up time recorded in weeks, to quantify the effect of booster dose administration, regression analysis was performed on the discrete-time hazard function of infection. Essentially, this quantity measures the risk of SARS-CoV-2 infection in each week for a HCW, conditional on not having contracted the infection until that week. Booster dose administration was specified as a categorical time-dependent covariate. Indeed, to be administered with the booster dose, a subject must have been free of infection until the time of administration. Therefore, the times of infection of the group of subjects receiving the booster dose were not directly comparable with those of the group which did not receive it, as the time origin was different between groups. 

According to a possible scenario of seasonal peaks in the conditional risk of SARS-CoV-2 infection during the follow-up time, a B-spline was utilized to smooth the possibly multipeaked baseline hazard function. A criterion of the minimization of the AIC was considered to choose the degree of its flexibility.

Also, to account for the within-HCWs difference in times from the completion of the two-dose vaccination cycle to the start of the follow-up, time from the completion of the two-dose vaccination cycle to the start of the follow-up was measured in weeks and regressed. This could be considered a proxy of the pre-existing level of immunity due to the vaccination cycle. Since the higher the time from the completion of the vaccination cycle to the start of the follow-up, the lower, potentially, the “booster” effect of the booster dose due to the lower pre-existing levels of the immune activity against the virus, the variable was regressed as having an interaction effect with the booster dose covariate. 

In addition, to account for infections that occurred before the follow-up, a simple categorical covariate was considered, with two categories: “previously infected”, “not previously infected”. To account for a possible differential effect of the booster dose for subjects previously infected and subjects not previously infected, in this case, an interaction effect with the booster dose administration covariate was also specified in the regression model. 

A sub-analysis of the NH1 data with complete information about demographic characteristics such as gender, age and occupation of the HCWs was performed to assess the effect of these other covariates on the hazard of infection. The same modeling procedure used for the whole cohort of HCWs was considered. In addition, to model a possible non-linear effect of the variable of age, a restricted cubic spline was utilized.

The exponentiated coefficients of the respective models related to the covariates offered the odds ratio (OR) of being infected with SARS-CoV-2 between covariate categories. For the sub-analysis, a forest plot represented the effect of the demographic categorical covariates, whereas the continuous non-linear effect of age was shown in a separate plot. 

All the statistical analyses were performed with R software (version 4.2.2). The function Lexis from the package Epi was utilized to augment the data to fit the discrete-time hazard model. 

## 3. Results

The databases of the three hospital networks exhibited great heterogeneity due to the difference in the data collected. Despite a relative uniformity of the demographic data such as gender, the heterogeneity chiefly concerned occupational data or the ways in which health data such as signs and symptoms were collected. One major challenge in retrieving data for the analysis was the lack of information about the demographic and occupational characteristics of all the HCWs. Namely, demographic information (age, occupation) about uninfected HCWs were not available for all the hospitals. To retrieve homogeneous information from the three HN and align it with the study objectives, a total cohort of 12,141 HCWs was considered. These individuals had complete information about the vaccination status (subjects must have completed the vaccination cycle) and time to an eventual infection of SARS-CoV-2 within the follow-up period was considered. Considering that usually, a period of 14 days is required from dose administration to complete the immunization process, subjects infected within 14 days from the booster dose administration were not included in the analysis.

Among the analyzed cohort of HCWs, 560 (5%) did not receive the booster dose within the specified time frame. In addition, 2563 (21%) HCWs became infected with SARS-CoV-2. The median time from completing the vaccination cycle to the start of the follow-up period was 34 weeks, with an interquartile range (IQR) of 4 weeks. 

As expected, the weekly risk of SARS-CoV-2 infection during the follow-up exhibited a distinct multi-peaked shape, with a major peak of the risk of infection at 15 weeks from the start of the booster campaign (January–February 2022) and a minor peak at 25 weeks (April 2022).

Unexpectedly, in univariable analysis, receiving the booster dose was slightly associated to higher risk of infection with respect to not receiving it, OR = 1.18, 95% confidence interval (CI) [0.99, 01.41]. This effect was counterintuitive since the booster dose is expected to provide greater protection against the infection. However, in the multivariable adjusted analysis, the booster dose was found to have a strong protective effect against infection, OR = 0.43, 95% CI [0.26, 0.74]. Notably, the interaction effect between the time elapsed since completion of the vaccine cycle and the administration of the booster dose was highly significant when tested with the Wald statistic (*p* < 0.001).

The modulating effect on the effect of the booster dose was both quantitative and qualitative, leading to the conclusion that it was non-protective against infection when the comparison was made between subjects with longer time since completion of the two-dose vaccine cycle. 

Regarding previous infections, the presence of an infection before the start of the follow-up was highly associated to the risk of infection, OR = 13.2 95% CI [8.00, 21.8]. Even though a previous infection might be associated to an increased protective effect of the booster dose, the differential effect of the booster dose in reinfected and not reinfected HCWs did not reach statistical significance. 

The results are clearly visualized in Figure 1, which shows, in three different panels, that the weekly infection risk profiles vary between subjects who received the booster dose (purple) and those who did not (blue) as a function of follow-up time (Time). The closer the completion of the vaccination cycle to the start of follow-up, the lower the infection risk profile of HCWs to whom the booster dose was administered compared to that of HCWs to whom it was not administered (Panel a). However, the longer the time since the completion of the vaccination cycle, the less obvious the booster effect of the booster dose in protecting against SARS-CoV-2 (Panels b and c). In particular, starting 25 weeks after completion of the vaccine cycle, the risk profiles cross over, leading the booster administration to be a (paradoxical) risk factor for infection.

### HN1 Sub-Analysis

Among the HN1 HCWs analyzed, 168 (3%) did not receive the booster dose in the time frame considered. In addition, 1424 (27%) became infected with SARS-CoV-2. The time from the completion of the vaccination cycle to the start of the follow-up had a median (IQR) of 34 (5) weeks. Table 1 describes the demographic and occupational characteristics of the HN1 HCWs.

The multipeaked shape of the conditional weekly risk of infection was like that found in the whole cohort.

The interaction between time from the completion of the vaccination cycle and booster dose administration was significant (*p* = 0.0014). Adjusted for the modulating effect of the time from completion of the vaccination cycle, the protective effect of the booster dose was evident, OR = 0.45, 95% CI [0.22, 0.91]. As shown in the analysis before, the modulating effect was both quantitative and qualitative, as the longer the time from the completion of the vaccination cycle, the less evident the protective effect of booster dose administration, leading to a flip in the risk profiles (Figure 1). Also, in this case, previous infection was highly associated to the risk of infection during follow-up. Figure 2 reports the ORs for sex, occupational characteristics and age. While sex is not associated to the risk of infection, age and occupational characteristics are. Indeed, compared to physicians, students and resident physicians display a significantly lower risk of infection. On the contrary, nurses display a higher risk of infection. In addition, age between 30 and 45 years is associated to higher risk of infection; on the other hand, age over 45 years is associated to a lower risk of infection.

## 4. Discussion

Following an initial analysis, our findings indicated that the population receiving a booster dose had a higher risk of infection than the population receiving only two doses. This finding may give rise to many debates about the rationale behind these data, which may support pseudoscientific beliefs and vaccine reluctance. In order to explain our preliminary results, we hypothesize that the increase in infections after booster dose administration could be linked to the timeframe in which the primary vaccination cycle was performed in relation to the time when the third dose was administered. For this reason, during our analysis, we assessed the timing between first or second dose and booster administration, observing that the booster dose’s time distance was a strict predictor of its protective effect. Indeed, some healthcare workers who became infected after receiving the booster dose were administered the dose several weeks following the proper date or twelve weeks following prior infection. Despite these results, the characteristics of our study’s design prevent us from identifying a causal relationship—only an association—between the timeframe between vaccination dose administration and the likelihood of infection. Consequently, it is possible that other factors influenced the probability of infection during the observation period. However, our findings highlight the importance of administering vaccines at the appropriate time and may apply to all vaccinations. Given the significance of immunization, the need for an early intervention, and the efficacy of correctly administered vaccines, the results of this study could be a cornerstone in the prevention and treatment of a future pandemic by increasing knowledge on the topic and aiding in the reduction in vaccination hesitancy among both healthcare workers and the general population.

In a recent study, Zoumpoulis et al. analyzed the attitude and acceptance of the booster doses of COVID-19 vaccination among physicians and discovered that the main reason for refusing vaccination was also the lack of information about the effectiveness of booster doses [15]. Another study (Della Polla, 2022) examined the extent to which HCWs intend to receive a booster dose of a COVID-19 vaccine and the factors that influence their willingness to accept it [16]. The authors highlighted that only 20,8% of HCWs totally agreed with the usefulness of the booster dose, while the main reasons for not getting vaccinated or for uncertainly were that the dose does not offer protection against the emerging variants and the fear of its side effects.

Since the introduction of COVID-19 vaccines, there have been multiple debates on the value of immunization in preventing infection and reducing the chance of reinfection. These discussions are frequently bolstered by theories that argue that natural immunity is preferable to immunity gained by vaccination. In 2022, Andrews et al. [17] conducted a study to estimate vaccine effectiveness against symptomatic disease caused by the Omicron and Delta (B.1.617.2) variants in England and highlighted that among the BNT162b2 primary course recipients, vaccine effectiveness increased to 67.2% (95% CI, 66.5 to 67.8) at 2 to 4 weeks after a BNT162b2 booster before declining to 45.7% (95% CI, 44.7 to 46.7) at 10 or more weeks. Other authors have made similar observations also considering data from reviews and meta-analyses; the authors include McMenamin, Zheng and Ssentongo [18,19,20]. Vishnoi et al. showed that among 2381 vaccinated HCWs, 4.4% reported post-vaccine COVID-19 infections. Out of the infected HCWs, only 0.9% reported severe cases, while 5.7% reported moderate disease severity [21].

In some cases, natural immunity results in a stronger immunity to the disease than that provided by vaccination; however, the dangers of this approach far outweigh the relative benefits [22], as in the case of COVID-19. Prevalence of 55 long-term effects [23] was evaluated, as well as that of consequent effects in terms of perceived work ability and fitness to work [24]. Some studies have investigated the influence of previous infection and vaccination in COVID-19 prevention considering the rate of reinfection or assessing the level of neutralizing antibodies after vaccination to evaluate the protective effect of vaccination [25,26,27].

Regarding the effectiveness of COVID-19 vaccine booster doses, Barda et al. suggested that a third dose of the BNT1622 mRNA vaccine is effective in protecting individuals from COVID-19 compared to receiving only two doses at least 5 months ago [28]. The authors showed a vaccine effectiveness of 93% for admission to hospital, 92% for severe disease and 81% for COVID-19-related death. 

The appearance of new variants, sometimes deeply different from the first strain spread worldwide, indeed could have impaired the vaccine’s effectiveness through small changes in the S-protein domain [29] that could lead to a mismatch between antibodies and the S-protein binding site of the new variants [30]. 

In a study performed by Chi, it was hypothesized that, while boosters effectively provided protection against severe illness and hospitalization caused by Delta, Omicron posed another challenge as the breakthrough infections increased drastically at the end of 2021 [31]. Fortunately, boosters were shown to increase serum anti-spike antibody levels and the neutralization titers against Omicron in recipients boosted with BNT162b2, mRNA-1273 or general mRNA vaccines. In addition, in the same study, CoronaVac and mRNA vaccine boosters were both shown to increase anti receptors binding domain-specific memory B cells and rapidly produce antibodies targeting diverse variants such as Omicron in boosted individuals. Although Delta and Omicron breakthrough infections can still occur in boosted individuals, viral loads seem to be lower, and symptoms are milder in these patients than in those that received two doses [31].

Following the COVID-19 epidemic, a number of authors have started to theorize about the next pandemic, when it may occur, and what might be done to avoid it [32,33,34,35,36,37,38,39,40]. The availability of effective vaccines since the first precedent makes vaccination readiness one of the primary topics [32]. 

Despite the important results mentioned above, our study suffers several limitations. The evaluation of the real-life effectiveness of the booster dose is a relevant aspect in a public health perspective, but not an easy task to perform. Indeed, the real-life setting essentially leads to the analyses of observational data derived from registries with all the issues related to observational studies: the risk of unmeasured confounding and the interpretation of the results only in an associative way. As already explained, in our study, the registries were different in the quantity and quality of information recorded, making it difficult to find common confounding variables that might affect the results related to the effect of booster dose administration, even though these were confirmed when considering some additional demographic and occupational characteristics in the sub-analysis. In addition, some positive NasoPharyngeal Swab (NPS) tests performed outside the hospital networks may have been missed when HCWs did not communicate their COVID-19 positivity to Occupational Health Physicians, or some vaccinations could be performed outside Lombardy. Finally, as reported, this was an observational study that could not identify a causal relationship between different times of vaccination and the risk of infection. In order to confirm these results in a causal rather than just associative manner, future research must measure confounding as much and as homogeneously as possible.

## 5. Conclusions

In conclusion, the COVID-19 pandemic has been a turning point in public health and epidemiology fields. It affected healthcare systems and organizations all over the world. COVID-19 also showed the importance of two main topics of concern in public health: preparedness and vaccinations. With regard to the latter, our study highlighted that the timeframe in the vaccine timetable and vaccine administration plays a key role. This element should lead to a stronger policy about vaccination and an increase in attention towards patients. This study could be considered as a further step in this topic, helping to increase attention to vaccination and preventive approach as well as the amount of knowledge in this specific field. 

## Figures and Tables

**Figure 1 vaccines-12-00315-f001:**
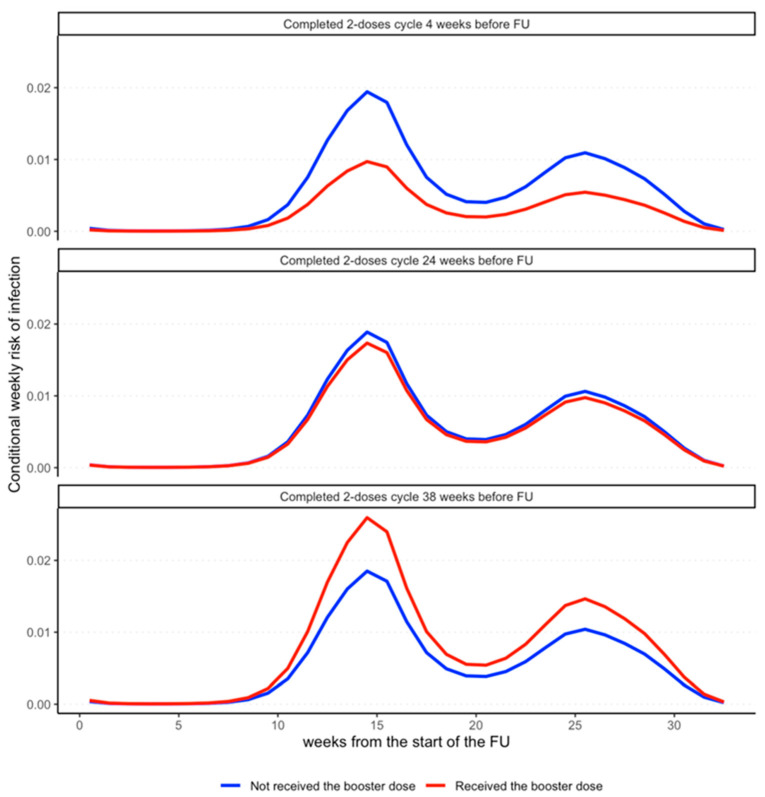
The differential effect of booster dose administration on the (conditional) weekly risk of SARS-CoV-2 infection. Weekly risk of infection for an HCW who was administered with the booster dose (red) and an HCW who was not (blue) form the start to the end of the follow-up period. The multipeaked shape of the risk profiles is due to the waves of infections characterizing the follow-up period of the study. The three sub-panels display the changes in the ratio of these risk profiles depending on the time from the completion of the two-dose vaccination cycle to booster dose administration. The protective effect of the booster dose is strong in those who enter the follow-up just after the completion of the two-dose vaccination cycle (4 weeks in the first panel), as the red profile has lower values compared to the blue. On the contrary, for those who completed the cycle far from the start of the follow-up, the effect of the booster dose increasingly vanishes (second panel), leading to a reversed effect in those who completed the two-dose cycle 38 weeks before entering the follow-up (third panel).

**Figure 2 vaccines-12-00315-f002:**
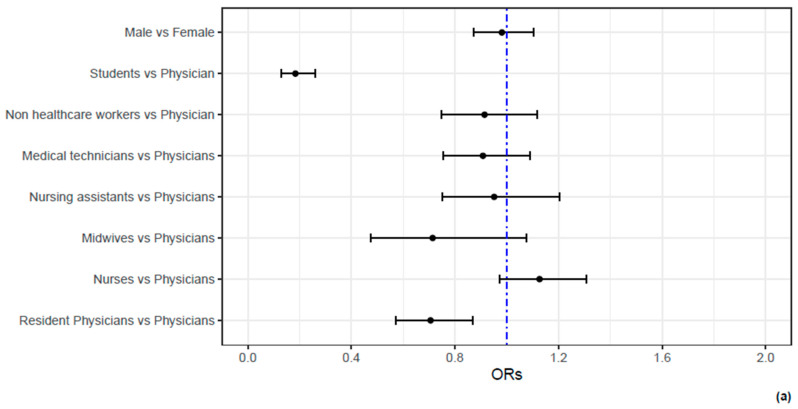
The association between demographic and occupational characteristics of HN1 HCWs and the odds of SARS-CoV-2 infection. In (**a**), the ORs and their 95% C.I.s for the demographic and occupational characteristics of HCWs are reported by means of a forest plot. It can be noticed that no difference in the odds of being infected are present between males and females. However, students and resident physicians display significantly lower odds of infection compared to physicians. In contrast, nurses display higher (even if not significant) odds of infection compared to physicians. In (**b**), a non-linear effect of age is reported. Compared to a hypothetical HCW of 50 years old, HCWs of 40 years old display a 10% increase in the odds of infection, whereas HCWs aged 60 years report a 20% decrease in the odds of infection.

**Table 1 vaccines-12-00315-t001:** Demographic and occupational characteristics of HN1 HCWs.

	Overall
Total Population	5142
Gender = M (%)	1530 (29.8)
Age (median [IQR])	38.00 [29.00, 52.00]
Occupation (%)	
Physicians	986 (19.2)
Resident physicians	685 (13.3)
Nurses	1222 (23.8)
Midwives	107 (2.1)
Nursing assistants	334 (6.5)
Medical technicians	645 (12.5)
Non healthcare workers	510 (9.9)
Students	653 (12.7)

## Data Availability

The data presented in this study are available on request from the corresponding author due to Privacy reasons.

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
