# Peer review of "The Effectiveness of the Third Dose of COVID-19 Vaccine: When Should It Be Performed?"

_vaccines, 2024, doi:10.3390/vaccines12030315_

Round 1

Reviewer 1 Report

Comments and Suggestions for Authors

This article investigates the efficacy of the third dose of COVID-19 vaccine. The results indicate that the correct vaccine plan and duration of use are crucial for the effectiveness of the vaccine. The results of the article contribute to the establishment of more effective vaccination strategies. However, there are some issues with the article:

1. What is the reason for the significant difference in the database performance of the three hospital networks?

2. The reason why the risk of infection is slightly higher when receiving booster immunity compared to not receiving booster immunity needs to be further explained in the discussion.

3. Can the vaccination survey of high-risk populations represent the overall research population?

4. Is the classification of high-risk workers in Table 1 reasonable?

5. The conclusion of the article does not correspond well to the title of the article.

Author Response

Manuscript ID: Vaccine-2885637

Title: The effectiveness of COVID-19 vaccine third dose: why it should be performed?

Milan, March 11ft, 2024

Dear Editor,

Dear Reviewer,

Thank you for your comments about our submission to Vaccines.

We carefully considered your precious suggestions and helpful notes, and revised the text accordingly, writing in red the changes made to the manuscript. A detailed list is enclosed.

We hope that the present version of MS will be suitable for publication in Vaccines.

Sincerely,

Author & co authors

Reviewer 1

Q1. What is the reason for the significant difference in the database performance of the three hospital networks?

A1. The difference can be explained by the difference in the health surveillance protocols implemented by the Occupational Medicine units of the various hospitals involved in the study, characterized by different demographic data collected.

Q2. The reason why the risk of infection is slightly higher when receiving booster immunity compared to not receiving booster immunity needs to be further explained in the discussion.

A2.

We have reformulated the discussion by trying to explain this aspect.

Q3.

A3. Our aim was not to make a study representative of the general population. Obviously health-care workera are on average younger and probably at greater risk of exposure than the general population.

Q4. Is the classification of high-risk workers in Table 1 reasonable?

A4. To add completeness of information regarding the study population, available demographic data collected from one of the hospitals involved in the study were reported.

Q5. The conclusion of the article does not correspond well to the title of the article.

A5.

We will change the title of the article by writing WHEN instead of WHY it should be performed.

Reviewer 2 Report

Comments and Suggestions for Authors

Dear authors,

Congratulations for your work!

I consider that the manuscript needs minor editing of English language. Just a few examples:

- line 153 - please remove “the“ before “before“

- line 174 - “signs“ instead of “sign“

- line 182 - “CoV2“ instead of “cov2“

Line 251 - 256 - a possible explanation for this differences would be interesting for the readers

Line 294 - “Assessed the infection rate in HCW after vaccination“ - the sentence is not finished

Comments on the Quality of English Language

See above.

Author Response

Manuscript ID: Vaccine-2885637

Title: The effectiveness of COVID-19 vaccine third dose: why it should be performed?

Milan, March 11ft, 2024

Dear Editor,

Dear Reviewer,

Thank you for your comments about our submission to Vaccines.

We carefully considered your precious suggestions and helpful notes, and revised the text accordingly, writing in red the changes made to the manuscript. A detailed list is enclosed.

We hope that the present version of MS will be suitable for publication in Vaccines.

Sincerely,

Author & co authors

Reviewer 2

Q1. I consider that the manuscript needs minor editing of English language. Just a few examples:
- line 153 - please remove “the“ before “before“
- line 174 - “signs“ instead of “sign“
- line 182 - “CoV2“ instead of “cov2“
- line 251 - 256 - a possible explanation for this differences would be interesting for the readers
- line 294 - “Assessed the infection rate in HCW after vaccination“ - the sentence is not finished

A1. Thanks for your suggestions, we revised these and other parts in the text.

Regarding lines 251-256, we added a possible explanation of the observed difference.

As for the sentence on line 294, that was the title of the subsection of the discussion. We removed it.

Reviewer 3 Report

Comments and Suggestions for Authors

Author Response

Manuscript ID: Vaccine-2885637

Title: The effectiveness of COVID-19 vaccine third dose: why it should be performed?

Milan, March 11ft, 2024

Dear Editor,

Dear Reviewer,

Thank you for your comments about our submission to Vaccines.

We carefully considered your precious suggestions and helpful notes, and revised the text accordingly, writing in red the changes made to the manuscript. A detailed list is enclosed.

We hope that the present version of MS will be suitable for publication in Vaccines.

Sincerely,

Author & co authors

Reviewer 3

Q1. One concern for me is that, I missed the information related on ethical approval for the study and secondly the absence of the acknowledgments to the occupational health units.

A1.  This is a study based on data collection for health surveillance purposes, therefore approval was not necessary for the processing of clinical data. We added the acknowledgements to the OH units.

Q2. Abstract:  Suggested to the authors to include the univariable and multivariable analysis with numbers to enhance the total picture of the manuscript and be more readable.

A2.  Done.

Q3. Introduction: Please delete the words (will be) lines 77 and 80.

A3. Done.

Q4. Results: Please reform the presentation of CI and OR (preferred separate) in line 206. OR, (0.45), 95% C.I, (0.22- 0.91). The same in line 246……………….

A4. Done.

Q5. Discussion:

  1. Please reform the discussion. The main purpose of the Discussion is to describe what your results mean and to compare with other studies. Your results presented for the first time on discussion section at the 4rd paragraph 191 line where you are referring about the aim of the study. I suggest enhancing the discussion paragraph debating the outcome of other researches detecting the vaccinations with the third or fourth dose.

A5a. Thanks for the helpful suggestion. We have adjusted the discussion paragraph's text in accordance with your feedback (pages 8-10)

  1. Add to the end of the discussion the main limitations of the study.

A5b. Done.

  1. Enhance the paragraph lines (294-303) with the follow reference: J Med Life. 2023 May; 16(5): 782–793. doi: 10.25122/jml-2023-0017.

A5c. Thanks for the suggestion. We added this reference.

  1. Please add to the section and discuss the follow references.
  2. Lancet. doi: 10.1016/S0140-6736(21)02249-2.
  3. Vaccines. 2023 Sep 12;11(9): 1480. doi: 10.3390/vaccines11091480
  4. Front Public Health. 2022 Dec 9; 10:1051035. doi: 10.3389/fpubh.2022.1051035.

A5d. Done.

Q6. References: Please delete the last reference (No 37) sounds to be a statement and not a reference

A6. Done.

Round 2

Reviewer 1 Report

Comments and Suggestions for Authors

The author answered many of my questions, and I am satisfied with them. The revised manuscript has greatly improved.

Reviewer 3 Report

Comments and Suggestions for Authors

Dear authors thank you for the revisions, the revised manuscript addresses all my concerns.